# Characterization of the Passion Fruit (*Passiflora edulis* Sim) bHLH Family in Fruit Development and Abiotic Stress and Functional Analysis of *PebHLH56* in Cold Stress

Yi Xu [1,2], Weidong Zhou [3], Funing Ma [1,2], Dongmei Huang [1], Wenting Xing [1], Bin Wu [1], Peiguang Sun [1], Di Chen [1], Binqiang Xu [1] and Shun Song [1,2,*]

1 State Key Laboratory of Biological Breeding for Tropical Crops, Haikou Experimental Station, Sanya Research Institute, Chinese Academy of Tropical Agricultural Sciences, Haikou 571101, China
2 Hainan Yazhou Bay Seed Laboratory, Sanya 572000, China
3 College of Agronomy and Biotechnology, Yunnan Agricultural University, Kunming 650201, China
* Correspondence: songs@catas.cn

**Abstract:** Abiotic stress is the focus of research on passion fruit characters because of its damage to the industry. Basic helix-loop-helix (bHLH) is one of the Transcription factors (TFs) which can act in an anti-abiotic stress role through diverse biological processes. However, no systemic analysis of the passion fruit bHLH (PebHLH) family was reported. In this study, 117 PebHLH members were identified from the genome of passion fruit, related to plant stress resistance and development by prediction of protein interaction. Furthermore, the transcriptome sequencing results showed that the *PebHLHs* responded to different abiotic stresses. At different ripening stages of passion fruit, the expression level of most *PebHLHs* in the immature stage (T1) was higher than that in the mature stage (T2 and T3). Eight *PebHLHs* with differentially expressed under different stress treatments and different ripening stages were selected and verified by qRT-PCR. In this research, the expression of one member, *PebHLH56,* was induced under cold stress. Further, the promoter of *PebHLH56* was fused to β-Galactosidase (GUS) to generate the expression vector that was transformed into *Arabidopsis*. It showed that *PebHLH56* could significantly respond to cold stress. This study provided new insights into the regulatory functions of *PebHLH* genes during fruit maturity stages and abiotic stress, thereby improving the understanding of the characteristics and evolution of the *PebHLH* gene family.

**Keywords:** bHLH; passion fruit; abiotic stress; cold; genome-wide analysis

## 1. Introduction

Transcription factors (TFs) play important roles in regulating growth and responding to external environmental stress in plants [1,2], which can regulate gene expression by binding to cis-promoter elements, thereby exerting regulatory roles in morphogenesis and so on [3,4]. TF genes, such as bHLH and Myeloblastosis (MYB), account for a large proportion of almost all plant genomes and are widely involved in plant development, stress response, and other physiological processes by regulating their target gene [5,6].

Basic/helix-loop-helix (bHLH) is a ubiquitous transcription factor family [7], which forms the second largest TF superfamily in plants [8]. The bHLHs have highly conserved alkaline/helix-loop-helix domains with approximately 50–60 amino acid residues [9]. This domain contains two functional regions: the basic region and the helix-loop-helix (HLH) region. The basic region contains approximately 15 amino acids and is located at the N-terminus, which is a critical region for binding to the cis-acting element E-box (5′-CANNTG-3′) and determining whether the bHLH transcription factor binds to the promoter region of genes [10,11]. The HLH region contains two α-helices connected by a relatively poorly

conserved loop distributed at the C-terminus, and this structure is essential for bHLH transcription factors to form homologous or heterodimers [10,12,13].

With the release of more high-quality genomes, many bHLH families in plants have been identified, such as *Carthamus tinctorius* (41) [14], Chinese jujube (92) [15], pineapple (121) [16], pepper (122) [17], potato (124) [18], peanut (132) [19], Jilin ginseng (137) [20], Brachypodium distachyon (146) [21], common bean (155) [22], tomato (159) [23], rice (167) [24], apple (188) [25], maize (208) [26], Chinese cabbage (230) [27], MOSO bamboo (448) [28], wheat (571) [29].

In current research, bHLHs are involved in responding to light [30], hormone signals [31], regulating anthocyanin biosynthesis [32], epidermal cell fate determination [33], and seed germination [34]. At present, the relationship between the bHLH gene and abiotic stress has attracted more and more attention. The bHLH has been shown to play a crucial role in plant resistance to abiotic stresses, such as abnormal temperature, drought, and high salinity. *FtbHLH3* of Tartary buckwheat participates in abiotic stress in response to changes in the polyethylene glycol (PEG) and the abscisic acid (ABA) [35], *FtbHLH2* can improve cold tolerance in plants [36]. *TabHLH39* augments the tolerance of transgenic *Arabidopsis* seedlings to drought, high salt, and low-temperature stress [37]. Overexpression of *AtbHLH92* can significantly improve salt and drought tolerance [38]. The *AtICE1/2* in *Arabidopsis* and their homologs *PtrbHLH* in Poncirus trifoliata can adjust plant cold tolerance and regulate peroxidase to break down hydrogen peroxide [39].

In addition, some studies have reported that *bHLH* genes are related to fruit development. In transgenic tomatoes, *SlbHLH22* is highly induced with the fruit color changed from green to orange, which is achieved by promoting the production of ethylene. Meanwhile, *SlbHLH22* was upregulated by using the exogenous ACC, IAA, ABA, and ethephon [40]. The research from Tan [41] showed that *CmbHLH32* was highly expressed in early developmental fruits. Overexpression of *CmbHLH32* can promote the early ripening of melon fruits, and the transgenic melons ripened earlier than the wild type. Papaya *CpbHLH1* and *CpbHLH2* promote fruit development by regulating genes related to carotenoid biosynthesis [42].

Passion fruit is a rare tropical fruit of the *Passiflora* genus (Passifloraceae), native to South America [43]. *Passiflora* has fresh and ornamental types [44]. *Passiflora* is nutritious and contains more than 100 different fruits in the pulp. At present, the planting area of East Asian countries such as Vietnam and China is growing rapidly, with a growth cycle of 4–6 months [45]. Because of its rich flavor, it has become more and more popular. Similar to other fruit, drought, high salinity, and cold and high-temperature stress seriously affect the development of passion fruit, resulting in a decline in fruit quality and yield. Due to the unpredictability of global climate change in recent years, passion fruit growing areas have been frequently affected by cold injury, which has also caused huge economic losses, resulting in more than 30% yield reduction and fruit stunting. Chilling injury is the most difficult to predict and control among the four abiotic stresses (drought, salt, cold and high temperature). Therefore, the identification of functional genes related to stress resistance and their utilization for variety improvement is of great importance for passion fruit cultivation.

Here, we identified the PebHLH family members in the genome of passion fruit and analyzed the members' biological information. Moreover, the expression patterns of *PebHLH* members were obtained by transcriptome sequencing and qRT-PCR at fruit developmental stages and under typical abiotic stress. More importantly, the *PebHLH* genes that were highly expressed and significantly induced by abiotic stress (drought, high salt, cold, and high temperature) were selected and overexpressed in *Arabidopsis* for functional validation. This provided a good foundation and reserved important information for studying the resistance mechanism in passion fruit and utilizing it for genetic improvement.

## 2. Materials and Methods

### 2.1. Identification of bHLH Members in Passion Fruit

The genome sequences of passion fruit were obtained from Phytozome V12.1. The HMM file of the NAM domain (PF00011) was downloaded from the PFAM database. In addition, analysis was performed using the bHLH with the highest comparison value to identify credible *PebHLHs*. The identification of PebHLH proteins used two methods described above were integrated and resolved to remove redundancy. AtbHLH protein sequences were obtained from Plant TFDB software (http://planttfdb.gao-lab.org/ accessed on 15 June 2022). The full-length protein sequences of PebHLHs and AtbHLHs were aligned by online ClustalX2. Moreover, the phylogenetic tree was constructed using MEGA (Version 7.0) [46]. A bootstrap test of 1000 repetitions was performed. Finally, PebHLH protein motifs were achieved using the MEME tool to compare the common domain of *PebHLHs*. Through the above procedures, the PebHLH members were finally obtained.

### 2.2. Gene Structure, and Chromosomal Locations of PebHLHs

The molecular weight (MW), protein isoelectric point (PI), and molecular formulas of all PebHLH members were reckoned using the online ProtParam. NetPhos (Version 3.1 Server) was used to predict the protein of PebHLH phosphate sites. The WoLF PSORT was used to perform Subcellular localization prediction. The gene structure maps, phylogenetic trees, combinations of motifs and gene structures, visualization of chromosomal localization, and collinearity relationships were obtained using TBtools [47].

### 2.3. Cis-Acting Elements, Protein Interaction Network and Gene Collinearity of PebHLHs

All *PebHLH* gene transcription start site DNA sequences of the genomes of 2000 bp upstream were imported to PlantCARE and used to analyze the sequence of the promoter region [48]. The Orthovenn2 was used to analyze the orthologous pairs between PebHLHs and AtbHLHs. The regulatory networks of *PebHLHs* and other proteins were identified using the AraNet (Version 2.0). The STRING database and the predicted regulatory network were evinced by Cytoscape software. The PebHLH gene duplication events between different species were analyzed by Multicollinearity Scanning Toolkit (MCScanX) [47].

### 2.4. Plant Materials, Transcriptome Sequencing, RNA Isolation and Reverse Transcription, Heat Map and qRT-PCR

Healthy passion fruit seedlings about two months old (30 cm in height) in the soil were chosen. The seedlings were planted in incubators and treated with cold, high temperature, high salt, and drought stresses. The control was placed in incubators (28 °C, 70% relative humidity, 12 h light/12 h dark cycle, 200 μmol m$^{-2}$ s$^{-1}$ light intensity). In addition, after fruit ripening, the pericarp turns purplish red. The three stages (T1, T2, and T3) are the time of 7d before ripening, ripening, and 7d after ripening, respectively [45]. The experimental material consisted of three biological replicated samples. The plant materials were used for transcriptome sequencing. The expression data of *PebHLHs* at four stress and three fruit ripening stages are shown in Tables S2 and S4. TBtools software was used for transcriptional analysis of *PebHLHs*, whose Z-score normalized FPKM values were used to produce heat maps [47]. The plant RNA isolation kit was used in *Arabidopsis* transformed with pCAMBIA1304-*PebHLH56*. The Biomic Biotechnology company (Beijing, China) was entrusted with sequencing services. The Primer sequences of *PebHLHs* were designed using the Primer5 software. The expression of *PebHLHs* was detected by qRT-PCR analysis. Relative expression levels were calculated using the $2^{-\Delta\Delta Ct}$ method and normalized to the *PebHLHs*.

### 2.5. Cloning the Promoter of PebHLH56 and Vector Construction

A 2000 bp DNA sequence before the start codon of the PebHLH56 was amplified by PCR and cloned into the pMD19-T vector. The promoter of PebHLH56 was assessed by DNA-MAN.

Expression vectors were constructed to examine whether *PebHLH56* responds to cold stress. The *PebHLH56* promoter PCR fragment was cloned into the pCAMBIA1304 vector digested with NcoI/HinIII, which was called pCAMBIA1304-*PebHLH56p.* The vector was transferred into the EHA105 strain (*Agrobacterium*).

*2.6. Cold Stress Treatment of Transgenic Lines*

The Agrobacterium transformed with pCAMBIA1304-*PebHLH56*p were shaken at 28 °C in YEB medium containing Kan and Rif antibiotics, respectively, and added to an *Arabidopsis* transgenic infiltration solution (1/2 MS, 50 g/L) to OD 600 = 0.8–1.0. Inflorescence impregnation was used to transform *Arabidopsis thaliana*. Eight transgenic lines have been obtained. Then three plants from each of the two lines in the T2 generation were cultured for cold stress treatment [8]. The 14-day-old transgenic *Arabidopsis* was treated in an incubator at 4 °C for 0, 24, and 36 h, respectively [3].

*2.7. Detection of GUS Activity*

The transgenic *Arabidopsis* with pCAMBIA1304-*PebHLH56p* under normal and cold stress were GUS stained. For GUS staining, seedlings of two lines were incubated in X-Gluc solution for 24 h at 37 °C [45,49]. GUS enzyme activity was determined by 4-methylumbel ureylglucuronide fluorometry [50].

## 3. Data Analysis and Results

### 3.1. Identification of bHLH Family of Passion Fruit

There were 117 *PebHLH* members identified from the passion fruit genome by the methods of HMM, protein BLAST, and MEME analyses. The characteristics of the *PebHLH* (Table S1) were analyzed, and the length of the *PebHLH* CDS ranged from 276 bp of *PebHLH3* to 3006 bp of *PebHLH73*. The identified *PebHLHs* encoded proteins ranged from 91 amino acids of *PebHLH3* to 1001 amino acids of *PebHLH73*. In addition, all the *PebHLHs* were showed containing three phosphorylation sites (Tyr, Thr, and Ser). The subcellular location prediction showed that the PebHLHs were distributed in the periplasmic, cytoplasmic, extracellular, and outer membranes. The molecular weight ranged from 10.43 Da to 114.67 Da.

### 3.2. Evolutionary Analysis of the bHLH Genes

The phylogenetic tree was constructed using the full-length sequences of the PebHLH and AtbHLH proteins (Figure 1). The PebHLHs could be divided into eighteen groups according to the evolutionary relationship. In groups 1 to 18, the largest number of *PebHLHs* was in group 1, with 11 members, and the smallest number was two in group 9. The number of PebHLH members in the remaining groups was between 4–10. The known *AtbHLHs* were then used to infer the hypothetical homologous members of *PebHLHs*. Among group 1–6, *PebHLH11/101/80/31/42/88/98/41/18/108/54/46/13/66/44/14* exhibited the closest relationship with At3G50330.1, At3G19500.1, At2G31730.1, At2G43010.5, At1G73830.1, At1G05805.1, At1G51140.1, At4G29100.1, At2G43140.2, At1G27660.1, At1G29950.2, At5G65320.1, respectively. Among groups 7–12, *PebHLH85/112/62/102/38/97/92/100/47/61/19/52/45* was the most homogeneous gene of At4G30980.1, At4G00480.2, At5G51790.2, At2G22760.1, At2G31280, 1At3G23210.1, At2G16910.1, At5G57150.2, At1G32640.1, At2G24260.2, At4G36060.2, At3G19860.3, At5G56960.2, respectively. Among group 13–18, *PebHLH57/43/91/5/12/69/99/26/114/84/22/95*, was the best orthologous match of At1G18400.1, At4G21340.1, At5G08130.3, At3G24140.1, At1G71200.1, At4G30410.1, At1G25330.1, At5G62610.1, At2G46510.1, At5G50915.1, At5G61270.2, respectively.

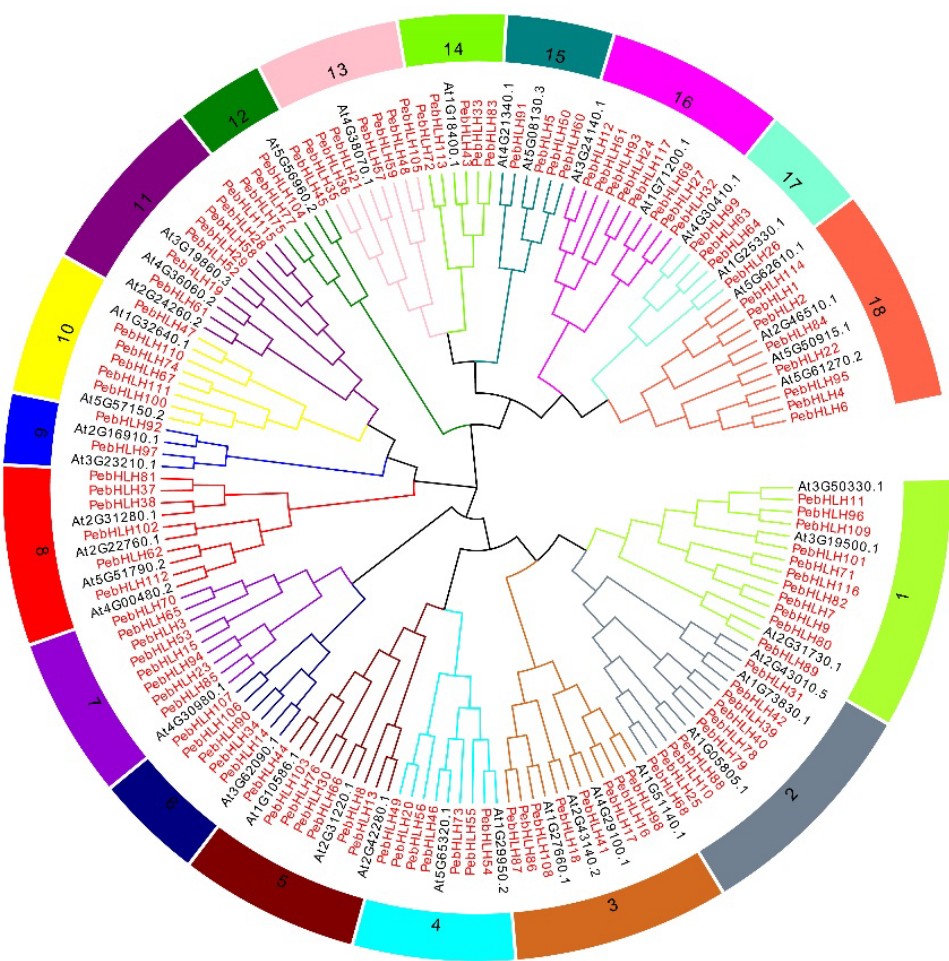

**Figure 1.** Phylogenetic relationship of bHLHs between passion fruit and *Arabidopsis*. The different colors of the outer circles represented groups 1–18. Pe and At are abbreviations of passion fruit and *Arabidopsis*, respectively.

### 3.3. Analysis of Gene Structure and Conserved Motifs

A total of 10 conserved motifs in *PebHLHs* were predicted (Figure 2). Although most members have two to four motifs, few members have only one motif; for example, *PebHLH26/48/91* only has motif one. *PebHLH30/38/41/43/52* only has motif two. Specifically, most members contain motifs 1 and 2, such as 111 members contain motif 1, and 114 members contain motif 2. In addition, 30 members contain motif 3, 38 members contain motif 4, 14 members contain motif 5, 17 members contain motif 6, and 4 members contain motif 7.

The exon and intron organization of *PebHLH* gene DNAs were analyzed. The bulk of members had the structure of 5′ and 3′UTR (untranslated region), and 18 only had exons and introns. Among them, nine members (*PebHLH14/27/31/48/55/81/95/98/111*) only had 5′UTR, and 23 members only had 3′UTR. The number of exons was between 2–11(Figure 2).

### 3.4. Chromosome Distribution of the PebHLHs

The chromosome distribution of each *PebHLH* was obtained, as shown in Figure 3. The total 117 *PebHLHs* were located on nine chromosomes. Seven members were located at unknown chromosomal positions. Among them, Chr1 contained the largest distribution of 41 *PebHLH* genes (~37.27%), followed by Chr6 (24 genes, ~21.82%). Meanwhile, Chr9 distributed the least number of *PebHLH* genes (3 genes, ~2.73%). Chr2, Chr3, Chr4, Chr5, Chr7, and Chr8 presented 7, 6, 10, 5, 4, and 10 *PebHLH* genes, respectively.

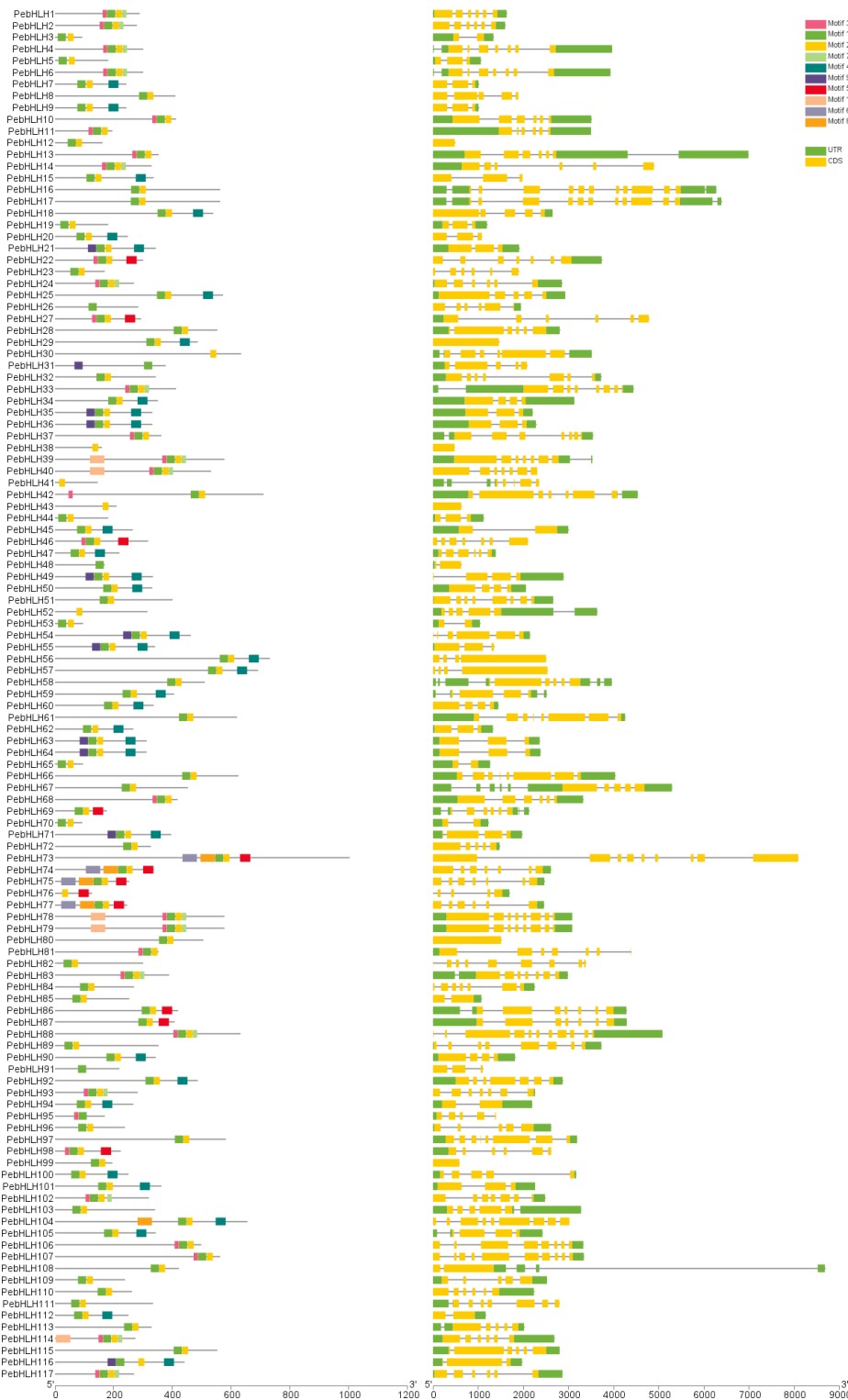

**Figure 2.** Phylogenetic clustering, conserved motifs, and exon/intron organization of *PebHLHs*. Ten color boxes indicate different motifs, green color represents 5′ and 3′UTR, and yellow color and black lines represent exons and introns, respectively.

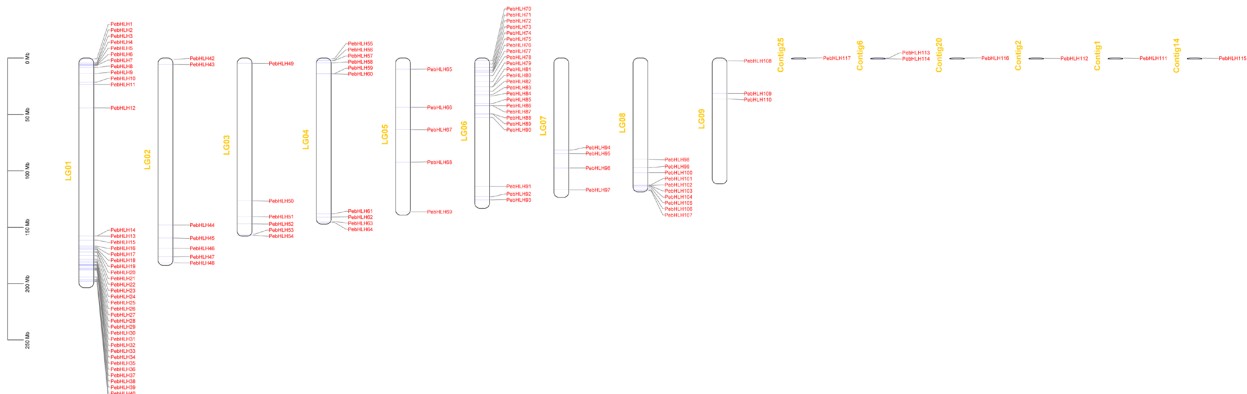

**Figure 3.** Locations of the 117 identified *PebHLHs* in 9 chromosomes of Passion fruit. The *PebHLHs* were located on No. 1, 2, 3, 4, 5, 6, 7, 8, and 9 chromosomes.

### 3.5. Promoter Analysis of PebHLH Genes

The TATA and CAAT boxes were the key cis-regulatory elements, which were found in all 117 *PebHLHs* (Figure 4). Additionally, the promoter region of the *PebHLH* family contains a large number of action elements related to abiotic stress, such as the cold-responsive element (CCGAAA) and the salicylic acid-responsive element (CCATCTTTTT and TCAGAA-GAGG). Some cis-acting elements are associated with stress response, such as the wound-responsive element (AAATTTCCT). In addition, there are some hormone-related elements, such as the abscisic acid responsiveness element (ABRE, GACACGTGGC, and ACGTG), the MeJA-responsive motifs (TGACG and CGTCA), the gibberellin-responsive motifs (CCTTTTG, TATCCCA, and TCTGTTG), and the auxin-responsive element (AACGAC).

### 3.6. Interaction Networks Analysis of PebHLHs

The protein interactions of PebHLHs were predicted using an orthogonal approach to further understand their biological functions. The results showed that 45 PebHLHs had an orthologous relationship with *Arabidopsis*, and 24 interacting proteins had been found. Most of the proteins that interacted with *PebHLHs* were WAKY, JAZ (Jasmonate ZIM-domain), COl (CONSTANS-Like), NINJA (Novel interactor of JA ZIM-domain), EPF (Early Pregnancy Factor), AOS (Allene Oxide Synthase), PFT (Pore-forming Toxin-like), CRY (Crystal protein gene), RBR (Retinoblastoma-related gene), ARF (Auxin response factor) TRY (Tryptophan) and so on (Figure 5). The function of the interacting proteins was related to the stress resistance and development of plants, which indicated that the function of PebHLH was also related to these aspects.

### 3.7. Collinearity Analysis of PebHLHs

All *PebHLH* genes were found to be displayed on chromosomes except for *PebHLH111-117*. There are 68 collinearity pairs that can be found. Most *PebHLHs* were found with one paralogous gene (Figure 6A).

The collinearity analysis among species was performed, and respectively, there were 12,406, 3118, and 25,224 genes in *Arabidopsis*, rice, and poplar, having a collinearity relationship with passion fruit. Among them, members of the PebHLH family had 91, 31, and 170 connections with *Arabidopsis*, rice, and poplar, respectively (Figure 6B). This result indicates that the homologous relationship between passion fruit and poplar is the closest, followed by *Arabidopsis* and rice.

Out of the 117 members, 27 and 22 genes had two and one homologous genes with the other three species, respectively. Other members have three to eight homologous genes. Among them, *PebHLH56* has 9 homologous genes (At1G01260.3, At1G63650.3, At2G46510.1, At4G00870.1, PNT03691, PNT03984, PNT45271, PNT50177, PNT53769) in collinear connections with *Arabidopsis* and poplar. In addition, 18 genes (*PebHLH27/28/33/35/36/40/46/58/61/ 67/78/91/97/98/99/109/110/115*) had one orthologous gene in rice.

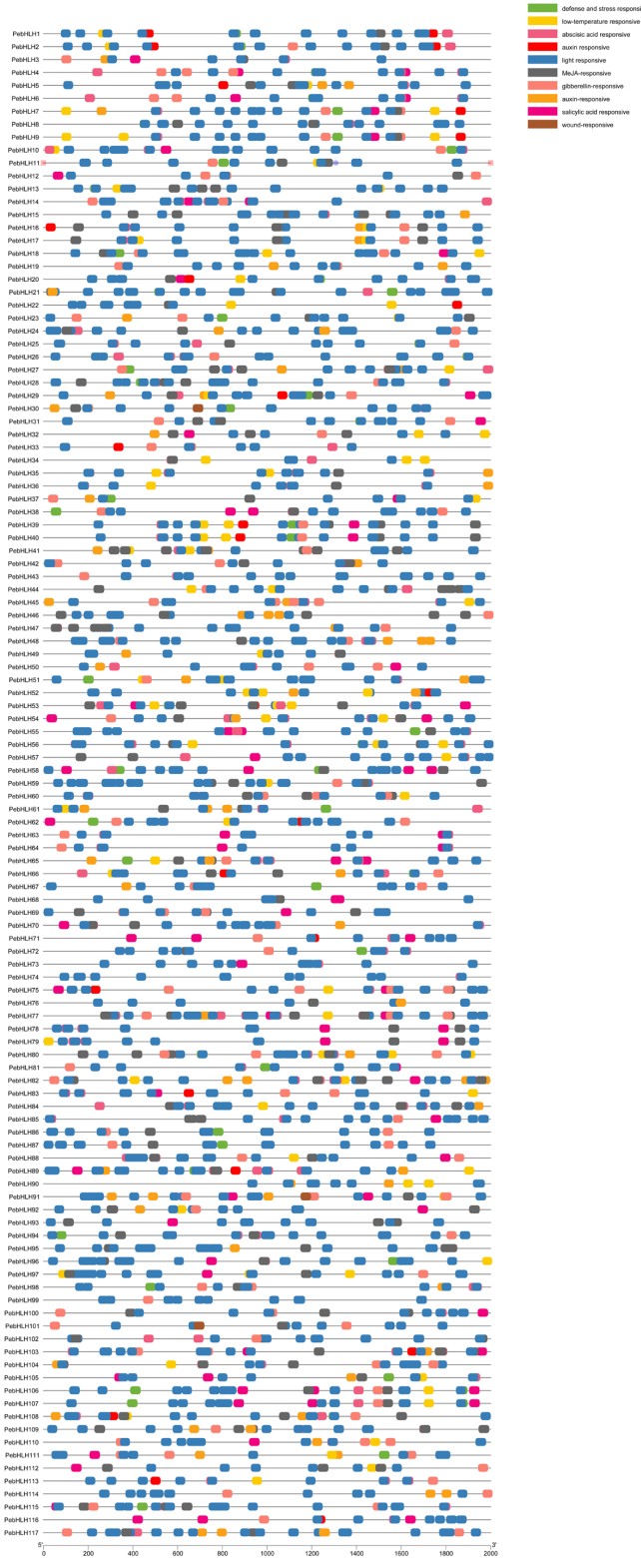

**Figure 4.** The cis-acting element analysis of 2000 bp promoter upstream of *PebHLH* genes. Functional descriptions of cis-acting elements were labeled by different colors.

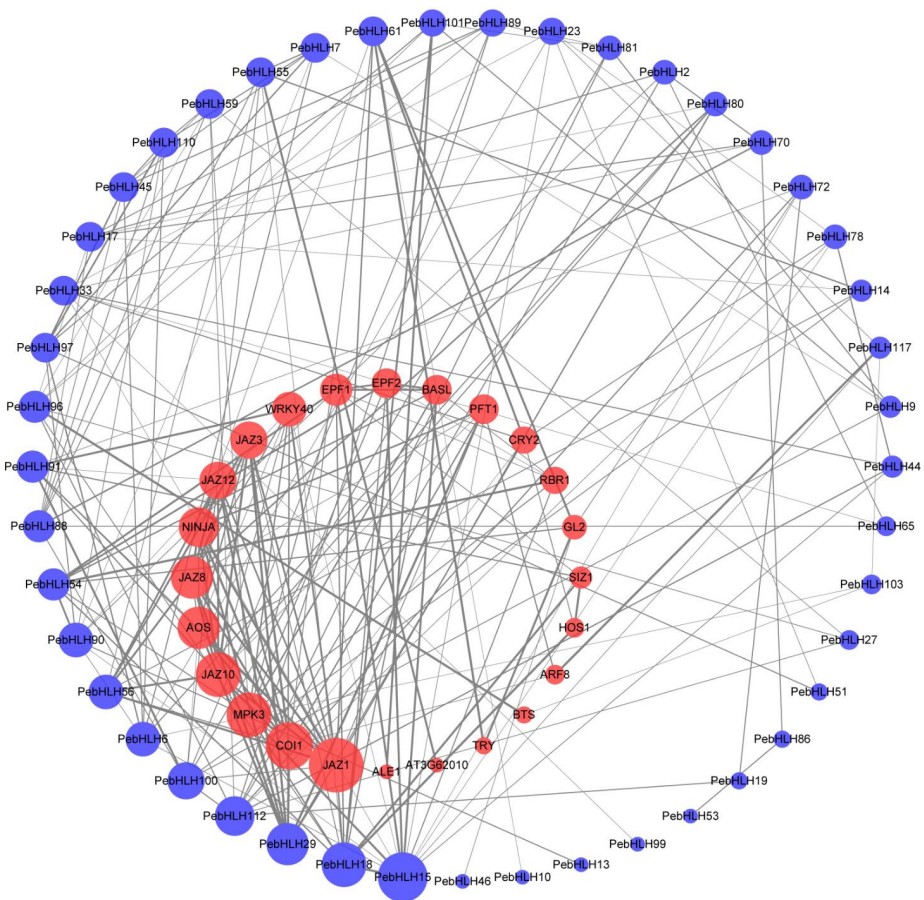

**Figure 5.** Protein interaction networks prediction of PebHLH. The outer blue circles indicate PebHLH proteins, and the inner red circles indicate other proteins in the reciprocal relationship. Interacting relationships are indicated by straight gray lines.

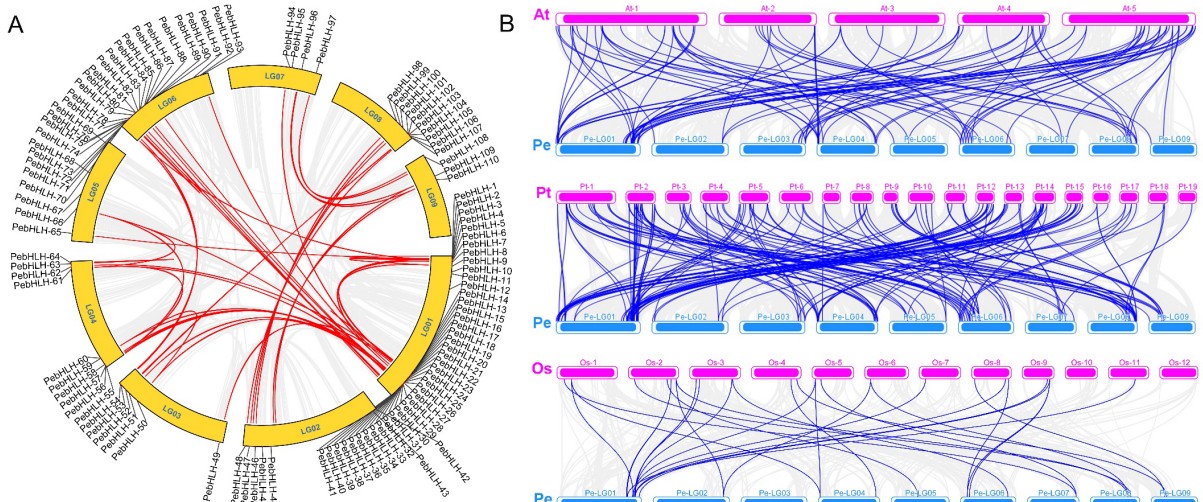

**Figure 6.** The synteny analysis of bHLHs of passion fruit genomes (**A**) and with other species (**B**). The red line represents the *PebHLH* gene pair, and the blue lines represent homologous gene pairs. Pe, Os, At, and Pt represent passion fruit, rice, *Arabidopsis*, and poplar, respectively.

### 3.8. Expression Analysis of PebHLHs

Transcriptome analyses showed that the PebHLH genes were differentially expressed under four abiotic stresses (Figure 7). *PebHLH10/11/25/35/36/43/46/68/69/80/117* were in-

duced by drought stress, while *PebHLH6/8/55/63/64/70/83* were suppressed. For the salt treatment, the expression of most genes was highest when treated with salt stress for 3 days. Especially, *PebHLH4* and *PebHLH56* were significantly upregulated under salt stress (NaCl 10d). However, the expression of *PebHLH8/55/63/64/70/83* was suppressed. Under high-temperature stress, some *PebHLHs* were upregulated, such as *PebHLH1/2/5/73/74/92/104*, and some genes were suppressed, such as *PebHLH8/55/63/64/70/83*. Under cold stress, the expression of the most gene had been induced, such as *PebHLH28/29/34/56/57/65/67/85/91/96/107/106/112*. We performed qRT-PCR validation analysis on some of the *PebHLH* genes (Figure 8). The result showed that the gene expression patterns were the same as heatmaps. The *PebHLHs* could respond to various abiotic stresses.

The expression levels of *PebHLHs* were obtained based on transcriptional sequencing results [51] at three different fruit ripening stages (T1, T2, and T3) (Figure 9). Most genes had the highest expression levels in the first period (T1) and regularly decreased in T2 and T3. And the expression of some genes was highest in T2, such as *PebHLH5/6/28/34/41/42/65/68/81/91/95/100/105/106/107*. A few reached the maximum expression in T3, such as *PebHLH13/17/52/62/69/73/74/76/85/101/103/104/109*. This result indicated that the expression of most *PebHLH* genes was negatively correlated with fruit ripening. Some genes were chosen to do the qRT-PCR verification (Figure 10), which showed that the expression trends of *PebHLHs* were consistent with the transcriptome.

*3.9. Response of Transgenic* Arabidopsis *to Cold Stress*

In A, GUS staining of the pCAMBIA1304-*PebHLH56p* transgenic seedlings was mainly observed in the leaf, and under the cold stress, GUS staining was enhanced and distributed throughout the transgenic plants. With the increase of cold time, GUS staining gradually deepened, and the GUS enzyme activity also increased (Figure 11).

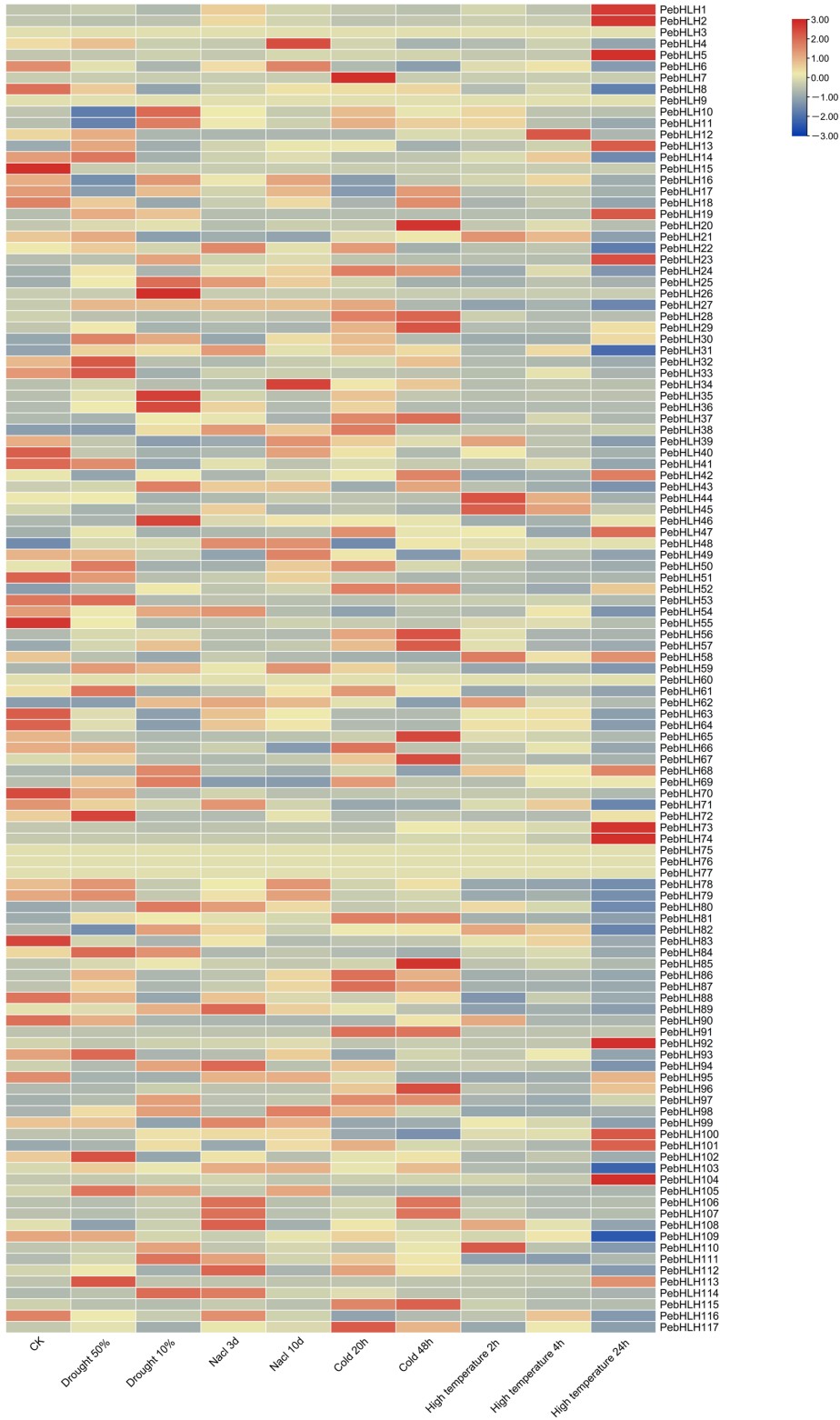

**Figure 7.** Differential expression of *PebHLHs* responding to different levels of drought, high salt, cold, and high temperature (Table S2). CK is normal growth condition, drought 50% and 10% are soil water content indexes, NaCl 3 d and 10 d are the treatment time under 300 mM concentration, cold 20 h, and 48 h are the treatment under 8 °C, and high temperature 2 h, 4 h, and 24 h are the treatment time under 42 °C, respectively.

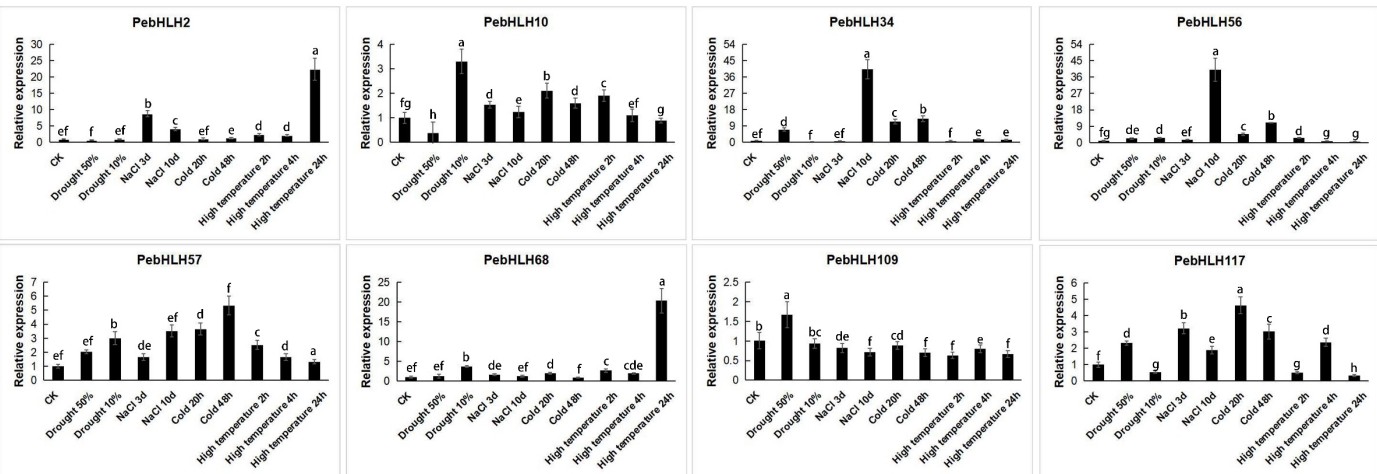

**Figure 8.** qRT-PCR analysis of 8 *PebHLHs* under 4 different abiotic stresses (Table S3). Horizontal coordinates indicate different stress treatments (the different processing methods are described above), and vertical coordinates represent relative expression values. The different letters mean significance, which was examined by Duncan's range test (*p* < 0.05).

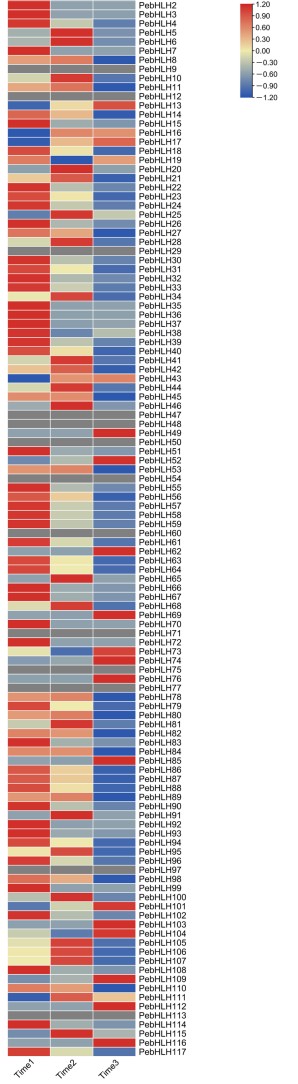

**Figure 9.** Differential expression of *PebHLHs* during 3 fruit ripening periods (Table S4).

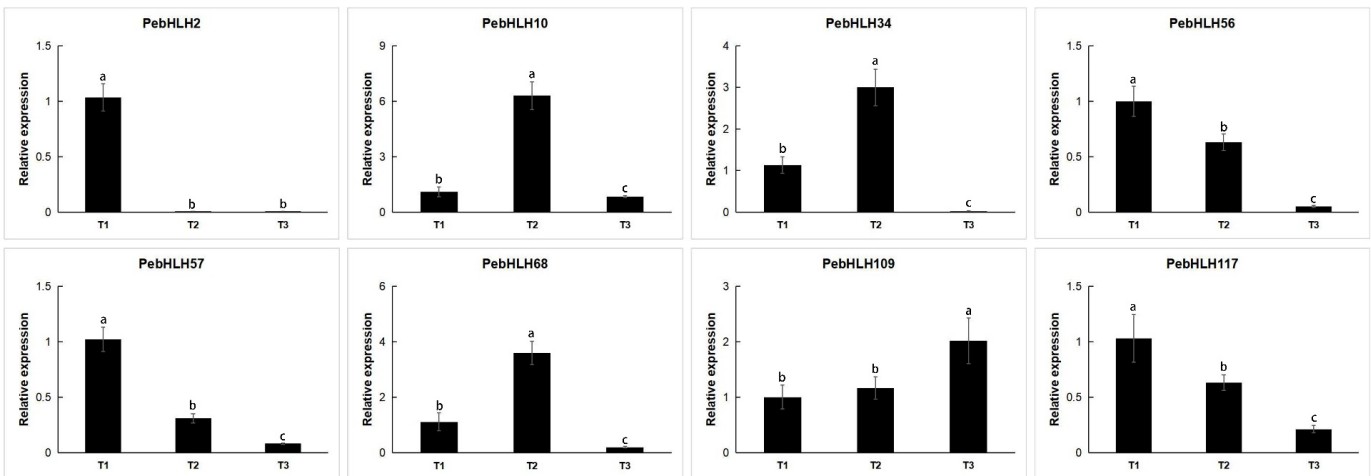

**Figure 10.** qRT-PCR analysis of 8 *PebHLHs* during 3 fruit ripening periods (Table S5). Horizontal coordinates indicate different stress treatments, and vertical coordinates represent relative expression values. Biological replicates, tests, and *p* values are described above. The different letters mean significance, which was examined by Duncan's range test ($p < 0.05$).

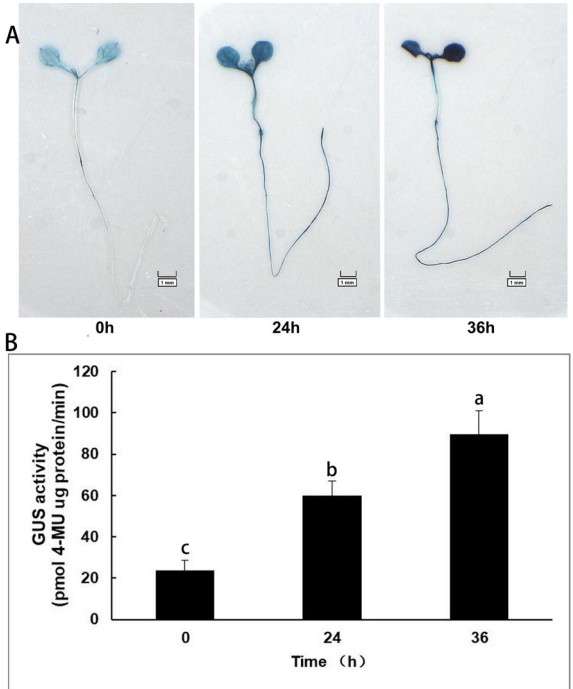

**Figure 11.** Induction and expression pattern of *PebHLH56* under cold stress. (**A**) GUS staining of overexpressing *Arabidopsis* strains. (**B**) GUS activity quantitative analysis of overexpressing *Arabidopsis*. Horizontal coordinates indicate different stress treatments, and vertical coordinates represent relative expression values. The different letters mean significance, which was examined by Duncan's range test ($p < 0.05$).

## 4. Discussion and Analysis

Numerous studies have shown that bHLH transcription factors are involved in diverse biological processes and the whole growth cycle [2]. At present, however, the systematic characterization of the bHLH genes in passion fruit is lacking, although bHLHs have been identified in many plants. The first plant bHLH gene was identified in maize (*Zea mays* L.) [52]. Furthermore, in this study, 117 bHLH genes were identified and characterized in passion fruit. This number is similar to some reported species, for example, the pineapple

(121) [53], the pepper (122) [28], the potato (124) [54], the Jilin ginseng (137) [55], and the tomato (159) [56]. The numbers were quite different from some species. For example, the MOSO bamboo was 448 [57], and the wheat was 571 [18]. This shows that the bHLH gene family is diverse in different species. Based on phylogenetic analysis, we classified 117 *PebHLH* genes into 18 subfamilies, which is the same number reported in maize [26]. Previous studies in other species reported six subfamilies in the animal genomes [55], while in plants, the bHLH family genes were divided into 21 in *Arabidopsis*, tomato, and pear [21,23]; 22 in rice [24] and 31 subfamilies in B. napus [53]. Compared with the classification of other plant species, our results show similarities and differences, which also indicate that the classification of bHLH transcription factors in plants is more complex than that in animals, so the classification of plant bHLH families is still needed for more research [58].

The gene structure of the *PebHLH* family was further analyzed. Most family genes contain two or more introns. Among them, *PebHLH12/29/99* have no intron, while seven members (*PebHLH3/45/48/53/65/70/94*) have only one intron. It is generally believed that genes with few or no introns in plants show lower expression levels [59]. The compact gene structure may facilitate the induction of gene expression in response to exogenous stress [60]. For example, intron-less genes, *PebHLH57*, were upregulated under drought, salt, cold, and heat stresses. Genome duplication events have occurred during plant evolution [61]. The evolution of the genome and the expansion of gene families depend primarily on gene duplication events [62]. We have performed the collinearity analysis on passion fruit, rice, and *Populus trichocarpa*, and the results showed that the *PebHLH* gene had the most tandem duplication relationship with the bHLH gene in *Populus trichocarpa*. Therefore, it is speculated that the relationship between passion fruit and *Populus trichocarpa* is the closest. In the same way, *PebHLH31* was found to be an ortholog of *AtPIF4* (AT2G43010.5). The AtPIF gene is a central signaling hub regulating plant growth and development [63].

The functions of bHLH genes in plants are diverse, including plant perception of the growth and development processes such as fruit development [24,64]. *AtCIB1*, together with *AtCRY*, promotes flower opening by stimulating the expression of flowering genes [65,66]. In our results, *PebHLH109* has the highest expression level in fruit, and it is predicted to play a key role in fruit development. It also shows the highest expression level at the T3 stage of fruit ripening. In pepper, *CabHLH33*, a homolog of *AtbHLH31*, was highly expressed in flower buds and petals. Previous studies suggest that *AtbHLH31* regulates petal growth by controlling cell expansion [67].

Recent studies have increasingly focused on the relationship between bHLH genes and abiotic stress. Under stress conditions, certain bHLH TFs are activated, and they combine with the promoters of key genes to regulate the transcription level of the target gene. Several studies have found that *OsbHLH068* of rice and At*bHLH112* of *Arabidopsis* play an active role in response to salt stress [68]. *MfbHLH38* of the *Myrothamnus flabellifolia* was transformed into *Arabidopsis*, and the drought tolerance of transgenic lines was enhanced by the increase of gene expression [69]. *ZmbHLH55* of Maize can increase salt stress tolerance by regulating the expression of ascorbic acid biosynthesis-related genes [70]. Overexpression of *Ntbhlh123* and *IbbHLH79* can improve the cold tolerance of tobacco [71] and sweet potato, respectively [72]. *ZjbHLH076/ZjICE1* of *Zoysia japonica* can enhance the tolerance of transgenic lines to cold stress [73]. The activity of gene expression is quantitatively efficiently regulated by specific or functional promoters that contain multiple cis-acting elements. These acting elements can respond to a variety of stress responses [52,74]. Plant bHLH responds to various stress responses through the activation of long terminal repeat (LTR) reverse transcription transposons [75,76], with activation factors including drought [77], heat [78], and salt [79]. In this work, the cis-element analysis indicated that *PebHLHs* contained elements (such as cold-responsive element, salicylic acid-responsive element, ABRE) that could be responsive to various stresses, which was consistent with previous studies on potato [18], lotus [80], and Pepper [56] bHLHs.

In this experiment, we focused on the function of the bHLH gene under cold stress. Here, we identified one of the *PebHLHs* (*PebHLH56*), which can respond to cold stress in

transgenic *Arabidopsis*. This study provides experimental evidence that bHLH family genes in passion fruit respond to cold stress, and we will further study the resistance mechanism of bHLH genes.

## 5. Conclusions

The bHLH gene family plays an important role in improving plant response to stress and plant development. In this study, we identified 117 *PebHLHs* from the genome of the passion fruit, and the genomic information about PebHLH family members was analyzed. *PebHLHs* could respond to abiotic stresses, including drought, high salinity, cold and high-temperature stresses, and they were differentially expressed in different fruit developmental stages. We were concerned about a gene *PebHLH56* that was induced under cold stress. The pCAMBIA1304-*PebHLH56p* transgenic *Arabidopsis* had a significantly deeper GUS staining than the control under cold stress, and the GUS enzyme activity also increased. The RT-qPCR result showed that the expression of *PebHLH56* was induced by cold stress. This study provides new insights into the regulatory functions of *PebHLHs* during abiotic stress and fruit ripening and screens a stress-resistance gene as a candidate member to improve the understanding of *PebHLH* gene family characteristics and evolution.

**Supplementary Materials:** The following supporting information can be downloaded at: https://www.mdpi.com/article/10.3390/horticulturae9020272/s1, Table S1. Basic information of bHLH genes identified in passion fruit; Table S2. The transcriptome data of passion fruit bHLHs under the drought, salt, cold and high-temperature treatment; Table S3. qPCR data of passion fruit bHLHs under the drought, salt, cold and high-temperature treatment; Table S4. The transcriptome data of passion fruit in the three stages of fruit ripening; Table S5. qPCR data of passion fruit bHLHs in the three stages of fruit ripening; Table S6. A list of the oligo primers of PebHLHs used for qRT-PCR.

**Author Contributions:** Y.X., W.Z., F.M., B.W., D.H., B.X. and W.X. performed experiments, D.C., S.S. and P.S. analyzed the data; Y.X. and S.S. drafted the manuscript. All authors have read and agreed to the published version of the manuscript.

**Funding:** The work was sponsored by Hainan Provincial Natural Science Foundation (321RC1088), Project of Sanya Yazhou Bay Science and Technology City (SCKJ-JYRC-2022-84, SCKJ-JYRC-2022-93), and National Natural Science Foundation of China (32260737).

**Data Availability Statement:** Not applicable.

**Conflicts of Interest:** The authors declare no conflict of interest.

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
