# Peer review of "Characterization of the Passion Fruit (Passiflora edulis Sim) bHLH Family in Fruit Development and Abiotic Stress and Functional Analysis of PebHLH56 in Cold Stress"

_horticulturae, doi:10.3390/horticulturae9020272_

Round 1

Reviewer 1 Report (New Reviewer)

The manuscript entitled " Characterization of the Passion fruit (Passiflora edulis Sim) bHLH family in fruit development and abiotic stress, and functional analysis of PebHLH56 in cold stress" is based on an original research work and of mean innovative value. The authors have identified the basic/helix-loop-helix (bHLH) family members in passion fruit, and by bioinformatics and comparative analysis try to identify their function. Data were obtained by transcriptome sequencing and qRT-PCR at fruit developmental stages and under typical abiotic stresses such as drought, high salt, cold, and high temperature, shows complicated expression level of different group of PebHLH genes. Selected PebHLH56 gene which cold-induced was transformed into Arabidopsis and the functional validation was sown.

This work delivers many interesting results and can be a source of valuable information. However, do not clear why authors for qRT-PCR validation analysis selected some of the PebHLH genes and why PebHLH56 gene was selected for the functional analysis. Some other PebHLH genes also induced under cold stress. Moreover, there are some minor issues. Most critical one is inadequate references list.

Too many typewriting errors/mistakes in the manuscript.

The authors made shortcomings that must be corrected before the publication of this work:

1)    Abstract:

Page 1 - *the star transcription factors* - new term?! unclear

2)    Introduction:

Page 2 - *This domain contains two functional domains: the basic domain and the helix-loop-helix (HLH) domain.* The authors indicate domain inside domain although in the next sentence they call it *region*

Page 2 - *the relationship of bHLH gene in abiotic stress* - unclear

Page 3 - *the four abiotic stresses*, *in high quality genome*

3)    Materials and Methods:

Page 3 - *AtbHLH protein sequences was obtained from Ensembl Plants software.* - It unclear to me that how the sequences could be obtained from software?

Page 3 - *the member of PebHLH members were finally obtained.* - need to rewrite.

Page 4 - *The experimental material consisted of three biological replicates sampled in three.* - need to rewrite.

4)    Data Analysis and Results:

Page 12 – Treatment temperature should be shown exactly.

5)    Discussion and Analysis:

Page 15 - *It is generally believed that genes with few or no introns in plants show lower expression levels [64]. The compact gene structure may facilitate the induction of gene expression in response to exogenous stress [65].* - need to rewrite.

Author Response

Thank you for your review. According to the amendments and suggestions, we have corrected the corresponding content, as follows, please check.

1)    Abstract:

Page 1 - *the star transcription factors* - new term?! unclear

Thank you for your suggestion. The mistakes have been corrected in the abstract section.

2)    Introduction:

Page 2 - *This domain contains two functional domains: the basic domain and the helix-loop-helix (HLH) domain.* The authors indicate domain inside domain although in the next sentence they call it *region*

Thank you for  pointing out the error. The mistakes have been corrected in the introduction section.

Page 2 - *the relationship of bHLH gene in abiotic stress* - unclear

Thank you for  pointing out the error. The mistakes have been corrected in the introduction section.

Page 3 - *the four abiotic stresses*, *in high quality genome*

Thank you for your suggestion.drought, “the four abiotic stresses” means drought, salt, cold and high temperature stresses. The mistakes have been corrected accordingly.

3)    Materials and Methods:

Page 3 - *AtbHLH protein sequences was obtained from Ensembl Plants software.* - It unclear to me that how the sequences could be obtained from software?

Thank you for  pointing out the error. The AtbHLH protein sequences was obtained from Plant TFDB software(http://planttfdb.gao-lab.org/). The mistake has been corrected in Materials and Methods section.

Page 3 - *the member of PebHLH members were finally obtained.* - need to rewrite.

Thank you for  pointing out the error. The mistake has been corrected.

Page 4 - *The experimental material consisted of three biological replicates sampled in three.* - need to rewrite.

Thank you for  pointing out the error. The mistake has been corrected.

4)    Data Analysis and Results:

Page 12 – Treatment temperature should be shown exactly.

Thank you for your suggestion. The treatment temperature and the other three treatment condition of abiotic stress have been added in the Fig.7 legend.

5)    Discussion and Analysis:

Page 15 - *It is generally believed that genes with few or no introns in plants show lower expression levels [64]. The compact gene structure may facilitate the induction of gene expression in response to exogenous stress [65].* - need to rewrite.

Thank you for your suggestion. The mistake has been corrected in the  Discussion and Analysis section.

Reviewer 2 Report (New Reviewer)

Good manuscript!!

Author Response

Thank you for your comments.

Reviewer 3 Report (New Reviewer)

Comments and suggestions for Authors

Dear Authors

Characterization of the Passion fruit (Passiflora edulis Sim) bHLH family in fruit development and abiotic stress, and functional analysis of PebHLH56 in cold stress

The  subject is very interesting and fall within the scope of the journal sections. The experimental dataset undoubtedly are useful and constitutes scientific values. The presented manuscript deals with the current global problem.  The aim of this study was identified the PebHLH family members in high quality genome of passion fruit, and analyzed the members biological information.

General remarks

In order to increase the usefulness of the article, Authors must refer to the following points.

Additions should be made to increase the scientific value of the manuscript.

Abstract: GUS - explain the abbreviation on first use.

Keywords: important words should be added.

Introduction: MYB - explain the abbreviation on first use.

Subsection 2.4.: The test site and soil conditions should be completed.

Discussion: There are too many paragraphs. Some can be combined.

Specific comments

References 49, 50, 61 - no citation in the text of the manuscript.

Description Figure 1, 2, 5, 6, 8, 10, 11 - not bold.

Figure 1, 2, 3, 4, 5, 6, 7, 8, 9, 10 - readability should be increased (improved).

The manuscript must be corrected according to the editorial requirements of the publisher.

Author Response

Thank you for your review. According to the amendments and suggestions, we have corrected the corresponding content, as follows, please check.

1.Abstract: GUS - explain the abbreviation on first use.

Thank you for your suggestion. The corresponding content has been added in the abstract section.

2.Keywords: important words should be added.

Thank you for your suggestion. The keywords have been added.

3.Introduction: MYB - explain the abbreviation on first use.

Thank you for your suggestion. The full name of MYB has been added in the introduction section.

4.Subsection 2.4.: The test site and soil conditions should be completed.

Thank you for your suggestion. The content has been added in the section 2.4.

5.Discussion: There are too many paragraphs. Some can be combined.

Thank you for your suggestion. Some paragraphs have been combined.

6.References 49, 50, 61 - no citation in the text of the manuscript.

Thank you for  pointing out the error. The references have been rearranged.

7.Description Figure 1, 2, 5, 6, 8, 10, 11 - not bold.

Thank you for your suggestion. The fonts of the Figure 1, 2, 5, 6, 8, 10, 11 legend have been changed.

8.Figure 1, 2, 3, 4, 5, 6, 7, 8, 9, 10 - readability should be increased (improved).

Thank you for your suggestion. The analysis of the figures has been revised.

9.The manuscript must be corrected according to the editorial requirements of the publisher.

Thank you for your suggestion. We have made some adjustments to the format of the manuscript, if there are any errors please help us to point them out, thank you.

This manuscript is a resubmission of an earlier submission. The following is a list of the peer review reports and author responses from that submission.

Round 1

Reviewer 1 Report

This work is compelling and engaging, and the results are mentioned in the state of new knowledge. It is well-structured and quality work. I make minor comments.

Three maturation periods are mentioned, what is the age of the fruits (DAF)?

I suggest adding photographs of the fruits analyzed in the figures needed

In figure 9: heat map of gene expression profile:  the abundance of some genes is suddenly observed in stage 3, could this situation be related to the biosynthesis of maturation components, pigments, and even senescence? I suggest extending the analysis of gene expression.

There is an error in the numbering of the figures, figure 13 does not exist and in its place is figure 11. Check the numbering sequence of the figures

Author Response

This work is compelling and engaging, and the results are mentioned in the state of new knowledge. It is well-structured and quality work. I make minor comments.

Reply: Thanks for your advice. Your suggestions are very valuable and some of them have good guiding significance for our future research work. We have made modifications and explanations according to your suggestions. In addition, we have made other modifications to the full text. According to the requirements of the editor, the traces of modifications are retained. Please check.

  1. Three maturation periods are mentioned, what is the age of the fruits (DAF)?

Reply: Thanks for your advice. We did neglect the relevant instructions. We have added a description of three maturity periods in Method section(2.4 Plant Materials…). As follows: and after fruit ripening, the pericarp turns to purplish red. The three stages (T1,  T2 and T3) are the time of 7d before ripening, ripening, and 7d after ripening, respectively.

  1. I suggest adding photographs of the fruits analyzed in the figures needed

Reply: Thanks for your suggestion. I agree with your suggestion very much. Adding a picture of fruit in the picture will let readers get more information more intuitively. However, we already used the image of the fruit in our last article. Therefore, I have added a description of the fruit period to the method. Please consider our explanation.

  1. In figure 9: heat map of gene expression profile:  the abundance of some genes is suddenly observed in stage 3, could this situation be related to the biosynthesis of maturation components, pigments, and even senescence? I suggest extending the analysis of gene expression.

 Reply: Thank you for your advice, which is very far-sighted. The research on the function of mature components and pigment biosynthesis is a very interesting and valuable direction. As you said, some genes have down-regulated and up-regulated expression patterns in three periods, which is what we have combined with the main content of this paper: Important genes related to stress resistance and fruit ripening process. Therefore, we comprehensively considered the genes with significant expression characteristics during stress resistance and ripening process for RT-qPCR analysis, and overexpressed Arabidopsis thaliana, and found that its cold tolerance had a better performance.

In Figure 9, there are still some genes that are highly expressed at the third maturity stage. We did not choose to do RT-qPCR analysis, that is, whether the T3 stage is related to fruit senescence as you mentioned. This relates to the fruit development process of passion fruit. The third stage is 7 days after harvest, when there is a certain shrinkage in the skin, if seen from the skin performance, it should be aging. But in terms of fruit flavor and taste, it is the best state.

At present, passion fruit is a minority fruit tree in the tropics, and most of the international basic research work focuses on developing countries or poor areas, which leads to the relatively backward basic research (including cytology, developmental biology, etc.) on growth and development, especially fruit development. Therefore, we cannot make a definitive conclusion of that the fruit in the third stage is senescent or developmental (postripening type).

So please consider our explanation.

  1. There is an error in the numbering of the figures, figure 13 does not exist and in its place is figure 11. Check the numbering sequence of the figures

Reply: Thanks for your suggestion. We have corrected the error.

Reviewer 2 Report

Plant bHLH proteins represent a superfamily of transcription factors that can be classified into 26 subfamilies that are evolutionarily closely related and have distinct functions. The transcription factor bHLH is known to regulate the expression of CBF genes in response to cold stress. ICE2 also induces the expression of the MYC-like transcription factor bHLH, which activates the expression of the CBF1 gene.

In this paper, the authors have done a lot of work to study the characterization of the Passion fruit bHLH family in fruit development and abiotic stress. In addition, the authors studied functional analysis of PebHLH56 in cold stress. It has been shown that Transcription factors Basic helix-loop-helix (bHLH) can act an anti-abiotic stress role through diverse biological processes. 117 PebHLH members into 18 subfamilies were identified from the passion fruit genome, related to plant stress resistance. It was shown that PebHLH56 could significantly respond to cold stress.

      The authors write in the purpose of the work “More importantly, the PebHLH genes that were highly expressed and significantly induced by abiotic stress….” It is necessary to indicate which abiotic factors have been studied. Why is the emphasis in the title of the manuscript on cold stress?

2.1. Apparently should replace "EnsemblPlants software" with "Ensembl Plants software".

2.4. The authors write “Healthy passion fruit seedlings about 30 cm in height were chosen, the seedlings were planted in incubators and treated with cold, high temperature, high salt and drought stress [45].” Detailed information should be given on the conditions of action of abiotic factors (duration of treatment with a stressor, salt concentration, etc.).

The authors write "The expression data of PebHLHs at four stress and three fruit ripening stages are shown in Tables S2 and S4." It is not clear to the reader whether seedlings 30 cm long can bear fruit. Specify the age of the seedlings. What three stages of fruit ripening did the authors study? What do Time1, 2, 3 mean in the table?

2.6. Are you sure you were working with transformed Arabidopsis plants? How was the transformation of Arabidopsis with the pCAMBIA1304-PebHLH56 and pCAM-BIA1304-PebHLH56p vectors, as well as the expression of the inserted genes, confirmed?

3.1. The authors write "The subcellular location prediction showed the PebHLH genes were distributed in the periplasmic, cytoplasmic, extracellular and outer membrane." Apparently, we are talking about gene products.

3.9. Replace "As shown in Figure 13A" with "As shown in Figure 11A".

Page 16. The authors write “…we have focused on the function of bHLH gene under cold stress. Here, we identified one of the PebHLHs (PebHLH56), which can respond to cold stress in the transgenic Arabidopsis.” However, in Figure 8 (in contrast to Figure 11C), PebHLH56 is not activated in Passion. The authors should explain this.

Author Response

Q1   Plant bHLH proteins represent a superfamily of transcription factors that can be classified into 26 subfamilies that are evolutionarily closely related and have distinct functions. The transcription factor bHLH is known to regulate the expression of CBF genes in response to cold stress. ICE2 also induces the expression of the MYC-like transcription factor bHLH, which activates the expression of the CBF1 gene.

In this paper, the authors have done a lot of work to study the characterization of the Passion fruit bHLH family in fruit development and abiotic stress. In addition, the authors studied functional analysis of PebHLH56 in cold stress. It has been shown that Transcription factors Basic helix-loop-helix (bHLH) can act an anti-abiotic stress role through diverse biological processes. 117 PebHLH members into 18 subfamilies were identified from the passion fruit genome, related to plant stress resistance. It was shown that PebHLH56 could significantly respond to cold stress.

Reply: Thank you very much for your earnest attitude and careful review, which is of great help to the improvement of our article. We made a lot of revisions and explanations, and kept the revision track as required by the editor. In addition to completing the questions raised by reviewers, we also made some other modifications. Please check.

Q2   The authors write in the purpose of the work “More importantly, the PebHLH genes that were highly expressed and significantly induced by abiotic stress….” It is necessary to indicate which abiotic factors have been studied. Why is the emphasis in the title of the manuscript on cold stress?

Reply: Thanks for your suggestion. In the penultimate paragraph of the preface, we added the expressions of four kinds of abiotic stress, while in the second penultimate paragraph, we added the great loss caused by chilling injury, which is the most intractable abiotic stress description, as follows: Due to the unpredictability of global climate change in recent years, passion fruit growing areas have been frequently affected by cold injury in recent years, which has also caused huge economic losses, resulting in more than 30% yield reduction and fruit stunting. Chilling injury is the most difficult to predict and control among the four abiotic stresses.

According to the editor's requirements, the modification trace has been retained, please check.

Q3  2.1. Apparently should replace "EnsemblPlants software" with "Ensembl Plants software".

Reply: Thanks for your suggestion. We have corrected the error.

Q4  2.4. The authors write “Healthy passion fruit seedlings about 30 cm in height were chosen, the seedlings were planted in incubators and treated with cold, high temperature, high salt and drought stress [45].” Detailed information should be given on the conditions of action of abiotic factors (duration of treatment with a stressor, salt concentration, etc.).

The authors write "The expression data of PebHLHs at four stress and three fruit ripening stages are shown in Tables S2 and S4." It is not clear to the reader whether seedlings 30 cm long can bear fruit. Specify the age of the seedlings. What three stages of fruit ripening did the authors study? What do Time1, 2, 3 mean in the table?

Reply: Thanks for your suggestion. We have added related instructions in both places, please check the modification trace.

(1)  “2.4.Plant Materials....”. We modified to “Healthy passion fruit seedlings about 30cm in height (seeding stage) were chosen, the seedlings were planted in incubators and treated with cold, high temperature, high salt and drought stress. And after fruit ripening, the pericarp turns to purplish red. The three stages (T1, T2 and T3) are the time of 7d before ripening, ripening, and 7d after ripening, respectively.”

(2)  In the picture description of Figure 7 and Figure 8. "CK is normal growth condition, drought 50% and 10% are soil water content indexes, NaCl 3d and 10d are the treatment time under 300mM concentration, cold 20h and 48h are the treatment under 8℃, and high temperature 2h, 4h and 24h are the treatment time under 42℃, respectively."

Q5  2.6. Are you sure you were working with transformed Arabidopsis plants? How was the transformation of Arabidopsis with the pCAMBIA1304-PebHLH56 and pCAM-BIA1304-PebHLH56p vectors, as well as the expression of the inserted genes, confirmed?

Reply: Thanks for your advice. I don't understand your question very clearly. I wonder if the following explanation is reasonable, please consider it. pCAMBIA1304-PebHLH56 was constructed into vector 1304, infected by Agrobacterium Tumefaciens, and overexpressed Arabidopsis thaliana. pCAM-BIA1304-PebHLH56p replaces the 35S promoter of 1304 vector with the promoter of PebHLH56. The writing of the carrier is also standard. Therefore, the logic of the three pictures in FIG. 11 is as follows: Figure A and Figure B show the expression of the pCAM-BIA1304-PebHLH56p transformation in Arabidopsis Thaliana (represented by GUS staining depth), which also indicates that the promoter has cold stress-related motif (because the 35s promoter is a universal promoter). This is followed by Figure C, which shows the transgenic Arabidopsis Thaliana with overexpression of PebHLH56 gene. Under the stress condition, the overexpressed strains can significantly induce the expression (the differential expression ratio is about 3, 0.5 times). In conclusion, the PebHLH56 gene (including its promoter) is sensitive to cold stress and can be induced to be highly expressed. In addition, the methods and results were not written in such detail, because the transformation of Arabidopsis thaliana is the same as the construction vector, method description and result description, and the journal is required to check the repetition rate function, so most of the time it is not written in great detail.

Q6  3.1. The authors write "The subcellular location prediction showed the PebHLH genes were distributed in the periplasmic, cytoplasmic, extracellular and outer membrane." Apparently, we are talking about gene products.

Reply: Thank you very much for your reminding. It was a really elementary mistake. We have modified it. “The subcellular location prediction showed the PebHLHs were distributed in the periplasmic, cytoplasmic, extracellular and outer membrane.”

Q7  3.9. Replace "As shown in Figure 13A" with "As shown in Figure 11A".

Reply: Thanks for your suggestion. We have corrected the error.

Q8  Page 16. The authors write “…we have focused on the function of bHLH gene under cold stress. Here, we identified one of the PebHLHs (PebHLH56), which can respond to cold stress in the transgenic Arabidopsis.” However, in Figure 8 (in contrast to Figure 11C), PebHLH56 is not activated in Passion. The authors should explain this.

Reply: Thank you for your earnest attitude. In Figure 8, PebHLH56 was located at the last of the first row. The results showed that, compared with CK, it induced high expression under low temperature stress, which was consistent with the transcriptome results (Figure 7). Figure 11C shows that the expression level of PebHLH5 in transgenic Arabidopsis thaliana was significantly increased under low temperature stress. So the results are consistent in terms of the function of PebHLH5. It may be caused by the poor definition of the picture in Figure 8. Considering the layout space, the picture will be very clear if stretched. In addition, the original picture we uploaded is also very clear.

Reviewer 3 Report

The research is useful for understanding the activity of  transcription factor bHLH during abiotic stress and fruit maturation in plant.

The results are theoretical and practical significance

The experiment design is appropriate, the figures and tables are appropriate

The cited references are relevance and up to date

The presentation of the MS is not very logical.  Data analysis should not be included in the result and discussion sections. They need to be included in the methods

As the mentioned above, MS need to be revised based on reviewer's comments and suggestions before accepting for publication

Author Response

The research is useful for understanding the activity of  transcription factor bHLH during abiotic stress and fruit maturation in plant.

The results are theoretical and practical significance

The experiment design is appropriate, the figures and tables are appropriate

The cited references are relevance and up to date

The presentation of the MS is not very logical.  Data analysis should not be included in the result and discussion sections. They need to be included in the methods

As the mentioned above, MS need to be revised based on reviewer's comments and suggestions before accepting for publication

Reply: Thanks very much for your review. Your suggestions are very helpful to the improvement of our article. We have made a large number of changes in the methods section of the paper, including we have added the data analysis expression to the methods section, or deleted the method data description in the results and discussion (repeated expression with the methods section). At the same time, we have also made corresponding modifications in the results and discussion parts to achieve a better logic of our research work introduction.